# Antibacterial effects of *Kampo* products against pneumonia causative bacteria

Yukiko Akahori[1,2]*, Yusuke Hashimoto[2], Kenichi Shizuno[3], Mitsuaki Nagasawa[2]

**1** Department of Microbiology, The University of Tokyo, Tokyo, Japan, **2** Department of Medical Technology and Sciences, School of Health Sciences at Narita, International University of Health and Welfare, Chiba, Japan, **3** Department of Clinical Laboratory, Chiba Kaihin Municipal Hospital, Chiba, Japan

* y-akahori@m.u-tokyo.ac.jp

**Data Availability Statement:** All relevant data are within the manuscript and its Supporting information files.

**Funding:** KAKENHI Grant-in-Aid for Early-Career Scientists (21K15654) Takeda Science Foundation.

## Abstract

Community-acquired pneumonia is caused primarily by bacterial infection. For years, antibiotic treatment has been the standard of care for patients with bacterial pneumonia, although the emergence of antimicrobial-resistant strains is recognized as a global health issue. The traditional herbal medicine *Kampo* has a long history of clinical use and is relatively safe in treating various diseases. However, the antimicrobial effects of *Kampo* products against pneumonia-causative bacteria remain largely uncharacterized. In this study, we investigated the bacteriological efficacy of 11 *Kampo* products against bacteria commonly associated with pneumonia. Sho-saiko-To (9), Sho-seiryu-To (19), Chikujo-untan-To (91) and Shin'i-seihai-To (104) inhibited the growth of *S. pneumoniae* serotype 3, a highly virulent strain that causes severe pneumonia. Also, the growth of *S. pneumoniae* serotype 1, another highly virulent strain, was suppressed by treatment with Sho-saiko-To (9), Chikujo-untan-To (91), and Shin'i-seihai-To (104). Minimum inhibitory concentration (MIC) and minimum bactericidal concentration (MBC) against these strains ranged from 6.25–50 mg/mL and 12.5–25 mg/mL, respectively. Furthermore, Sho-saiko-To (9), Chikujo-untan-To (91), and Shin'i-seihai-To (104) suppressed the growth of antibiotic-resistant *S. pneumoniae* isolates. Additionally, Sho-saiko-To (9) and Shin'i-seihai-To (104) showed growth inhibition activity against *Staphylococcus aureus*, another causative agent for pneumonia, with MIC ranging from 6.25–12.5 mg/mL. These results suggest that some *Kampo* products have antimicrobial effects against *S. pneumoniae* and *S. aureus*, and that Sho-saiko-To (9) and Shin'i-seihai-To (104) are promising medicines for treating pneumonia caused by *S. pneumoniae* and *S. aureus* infection.

## Introduction

Community-acquired pneumonia (CAP) is a common infectious respiratory disease associated with high morbidity and mortality in infants, elderly, and immunocompromised individuals worldwide [1, 2]. It is primarily caused by various microbial pathogens, including bacteria, fungi, and viruses. Respiratory pathogens enter the body through inhaled droplets, and colonize the upper respiratory tract or oral cavity, where they adhere to mucosal surfaces.

The funders had no role in the study design, data collection, and analysis, the decision to publish, or the preparation of the manuscript.

**Competing interests:** The authors have declared that no competing interests exist.

Then, the pathogens move to the lower respiratory tract through aspiration or compromised barrier defenses and adhere to alveolar cells. This leads to lung inflammation and pneumonia symptoms such as cough, shortness of breath, and fever [3].

*Streptococcus pneumoniae* is a leading cause of CAP, associated with severe diseases such as meningitis and sepsis [4–6]. In standard medical therapy, antibiotic therapy has been considered the first-line treatment for pneumococcal infection because *S. pneumoniae* is generally susceptible to antibiotics such as penicillin and macrolides [7]. Notably, the emergence of antibiotic-resistant *S. pneumoniae* strains has become increasingly common worldwide [7–9]. Over the past 25 years, the rates of community-acquired infections by penicillin-resistant and quinolone-resistant pneumococci have been slightly decreased by appropriate prescription of antibiotics and increased coverage of vaccination in some countries, including Japan and the USA [10–12]. However, it is anticipated that the incidence of pneumococcal infections will not decrease due to several factors; Some strains become antibiotic-resistant even when the improved antibiotics are developed. While pneumococcal vaccines are quite effective, they do not provide coverage for all strains [13].

*Staphylococcus aureus* also stands as a frequent causative agent of severe CAP worldwide [14]. Evidently, the emergence of methicillin-resistant *S. aureus* (MRSA) is a growing concern because MRSA infection leads to critical illness and high mortality rates [15, 16]. Additionally, *Klebsiella pneumoniae* infections have raised the healthcare alarms [17]. *K. pneumoniae* is a predominant cause of hospital-acquired pneumonia (HAP), with the increasing prevalence of carbapenem-resistant strains causing a global concern [18].

A traditional Japanese herbal medicine, *Kampo*, has been approved based on its historical and clinical efficacy and has been prescribed for patients with a wide range of symptoms and diseases. These include autoimmune diseases, inflammatory diseases, allergies, and cancer [19–23]. Also, *Kampo* products have been used clinically to manage infectious diseases [24, 25]. Furthermore, previous experimental studies have reported the antimicrobial properties of *Kampo* products. Toki-Inshi (86) has bactericidal activity against *Staphylococcus* spp. [26]. Shin'i-seihai-To (104) directly suppresses the growth of serotype 19F *S. pneumoniae* [27]. Hainosankyuto (122) has a protective effect against cutaneous *S. pyogenes* infection in mice by promoting macrophage phagocytic activity through modulation of IL-12 and IFN-γ [28]. Mao-To (27) blocks the uncoating process of the influenza virus [29], and treatment with Mao-To (27) alleviates the symptoms of influenza virus-infected mice by promoting humoral immune response [30]. Kakkon-To (1) inhibits virus replication through the acidification of endosomes and lysosomes [31]. Also, seven *Kampo* products (Otsuji-to (3), Dai-saiko-to (8), Saiko-keishi-to (10), Saiko-keishi-kankyo-to (11), Saiko-ka-ryukotsu-borei-to (12), Keishi-ka-ryukotsu-borei-to (26), and Bofu-tsusho-san (62)) directly inhibit the growth of *Trichophyton rubrum* [32]. Nonetheless, the comprehensive antibacterial efficacy of *Kampo* products against pneumonia-causing bacteria remains largely uncharacterized.

In this study, we investigated the direct antibacterial effects of 11 *Kampo* products against *S. pneumoniae*, *S. aureus*, and *K. pneumoniae*. We showed that Sho-saiko-To (9), Chikujo-untan-To (91), and Shin'i-seihai-To (104) had growth inhibition activity against clinical *S. pneumoniae* isolates. Additionally, they had antibacterial effects against antibiotic-resistant *S. pneumoniae* strains. Furthermore, Sho-saiko-To (9) and Shin'i-seihai-To (104) exerted antibacterial effects against *S. aureus*, but not *K. pneumoniae*. These findings provide new therapeutic ideas for treating pneumonia caused by *S. pneumoniae* and *S. aureus*.

## Materials and methods

### Bacterial strains

*S. pneumoniae* serotype 3 was kindly provided by Prof. K. Kawakami, Tohoku University. Clinical strains of *S. pneumoniae* serotype 1, 6A, and 19A were isolated in Chiba Kaihin Municipal Hospital. *S. pneumoniae* strains are highly auxotroph and fragile [33]. *S. pneumoniae* strains were cultured in nutrient-rich Todd-Hewitt broth (Difco, Detroit, MI, USA) at 35°C with 5% $CO_2$ and harvested at the mid-log growth phase [34]. *S. pneumoniae* strains were stored at −80°C until use. *E. coli* ATCC 25922 and *K. pneumoniae* ATCC 700603 were harvested on Mueller-Hinton II Agar (BD, Franklin Lakes, NJ, USA). *S. aureus* ATCC 43300 was harvested on 5% sheep blood agar (Kyokuto, Tokyo, Japan). The bacteria were stored in a microbank (Iwaki, Tokyo, Japan) at −80°C until use.

### Kampo

*Kampo* products (Kakkon-To (1), Kakkon-To-ka-senkyu-shin'I (2), Sho-saiko-To (9), Saiko-keishi-To (10), Hange-koboku-To (16), Sho-seiryu-To (19), Mao-To (27), Hochu-ekki-To (41), Chikujo-untan-To (91), Goko-To (95), Shin'i-seihai-To (104)) were purchased from Tsumura (Tokyo, Japan). *Kampo* products were suspended in distilled water, filtered using a 70 μm cell strainer (BD, New Jersey, USA), and stored at −20°C until use.

### Growth inhibition assay

The growth inhibition activity of *Kampo* products against bacteria was determined using the diffusion method in accordance with the test guideline by Clinical and Laboratory Standards Institute (CLSI). A 0.5 McFarland suspension of bacteria was fully soaked with a sterile cotton swab, and inoculated on Mueller-Hinton II agar with or without 5% sheep blood (BD). A 7.0 mm cork borer was used to prepare the hole for adding drops of *Kampo* products. For analysis of *S. pneumoniae*, Mueller-Hinton II agar with 5% sheep blood plates was incubated at 35°C with 5% $CO_2$ overnight. For analysis of *E. coli* ATCC 25922, *S. aureus* ATCC 43300, and *K. pneumoniae* ATCC 700603, Mueller-Hinton II agar plates were incubated at 35°C overnight.

### Antibiotic resistance test

Resistance to tosufloxacin tosylate hydrate (TFLX), a form of fluoroquinolone, was measured using a TFLX disk (Eiken Chemical, Tokyo, Japan) according to the manufacturer's instructions. Briefly, Mueller-Hinton II Agar with 5% sheep blood were inoculated with a 0.5 McFarland suspension of bacteria. TFLX disk was put and incubated at 35°C with 5% $CO_2$ overnight before the measure of inhibitory zone diameter. A diameter less than 16 mm is determined as resistant, 17–21 mm as intermediate, and larger than 22 mm as susceptible [35].

### Minimum inhibitory concentration and minimum bactericidal concentration

The minimum inhibitory concentration (MIC) of antibiotics (penicillin, meropenem, erythromycin, cefotaxime, imipenem, clindamycin, and levofloxacin) against *S. pneumoniae* serotype 1, 3, 6A, and 19A were tested using Optopanel OP1 (Kyokuto) according to the manufacturer's protocol. The antibiotic resistance profiles of *S. pneumoniae* serotypes 1 and 3 were summarized in S1–S3 Tables. The MIC of *Kampo* products (Sho-saiko-To (9), Chikujo-untan-To (91), and Shin'i-seihai-To (104)) were determined according to the CLSI guidelines [36]. Briefly, a McFarland suspension of 1 of bacteria was inoculated into Mueller-Hinton broth

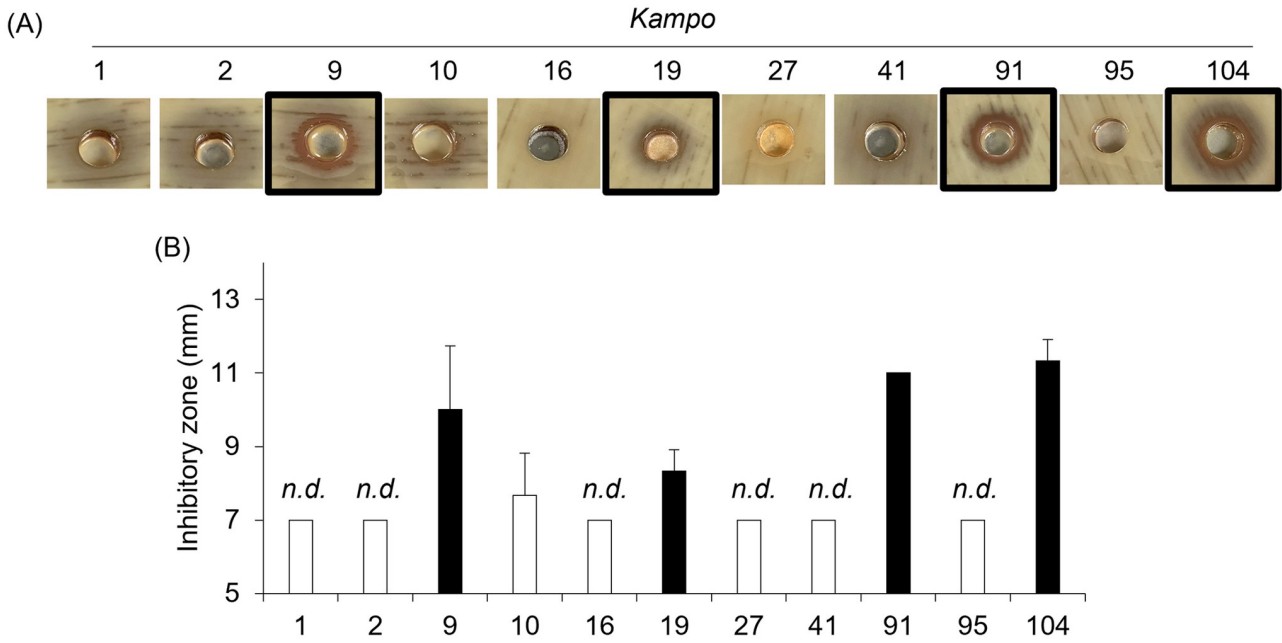

**Fig 1. The growth of *S. pneumoniae* serotype 3 is inhibited by treatment with Sho-saiko-To (9), Sho-seiryu-To (19), Chikujo-untan-To (91), and Shin'i-seihai-To (104).** *S. pneumoniae* serotype 3 (0.5 McFarland suspension) was incubated on Mueller-Hinton II Agar containing 5% sheep blood in the presence of each *Kampo* product at 35°C with 5% $CO_2$ overnight. (A) The representative photograph of the growth inhibition zone on Muller-Hinton II Agar plates. (B) The growth inhibition zone around the hole was measured. Data are mean ± SD. Data from three independent experiments were combined. The white bar indicates that no inhibition was observed. n.d.: not detected.

alone or containing 5% horse hemolysate (Kyokuto), followed by the addition of serially diluted *Kampo* products. The MIC represents the lowest concentration that could inhibit bacterial growth. Minimum bactericidal concentration (MBC) was determined by inoculating 10 μL of each well (from the MIC to higher *Kampo* concentration) in 5% sheep blood agar (Kyokuto). Plates were incubated as described in the grow inhibition assay section.

## Results

### The growth of *S. pneumoniae* serotype 3 is inhibited by treatment with Sho-saiko-To (9), Sho-seiryu-To (19), Chikujo-untan-To (91), and Shin'i-seihai-To (104)

We selected 11 *Kampo* products (Kakkon-To (1), Kakkon-To-ka-senkyu-shin'I (2), Sho-saiko-To (9), Saiko-keishi-To (10), Hange-koboku-To (16), Sho-seiryu-To (19), Mao-To (27), Hochu-ekki-To (41), Chikujo-untan-To (91), Goko-To (95) and Shin'i-seihai-To (104)) that have been frequently prescribed to patients with the common cold and investigated whether the 11 *Kampo* products have direct antibacterial effects against *S. pneumoniae* serotype 3. The diffusion assay revealed that the growth of *S. pneumoniae* serotype 3 was suppressed by treatment with Sho-saiko-To (9), Sho-seiryu-To (19), Chikujo-untan-To (91) and Shin'i-seihai-To (104) (Fig 1A and 1B).

### The growth of *S. pneumoniae* serotype 1 is inhibited by treatment with Sho-saiko-To (9), Chikujo-untan-To (91), and Shin'i-seihai-To (104)

We next examined the antibacterial effects of 11 *Kampo* products on *S. pneumoniae* serotype 1, another clinically isolated strain from a patient with invasive pneumococcal disease (Fig 2A

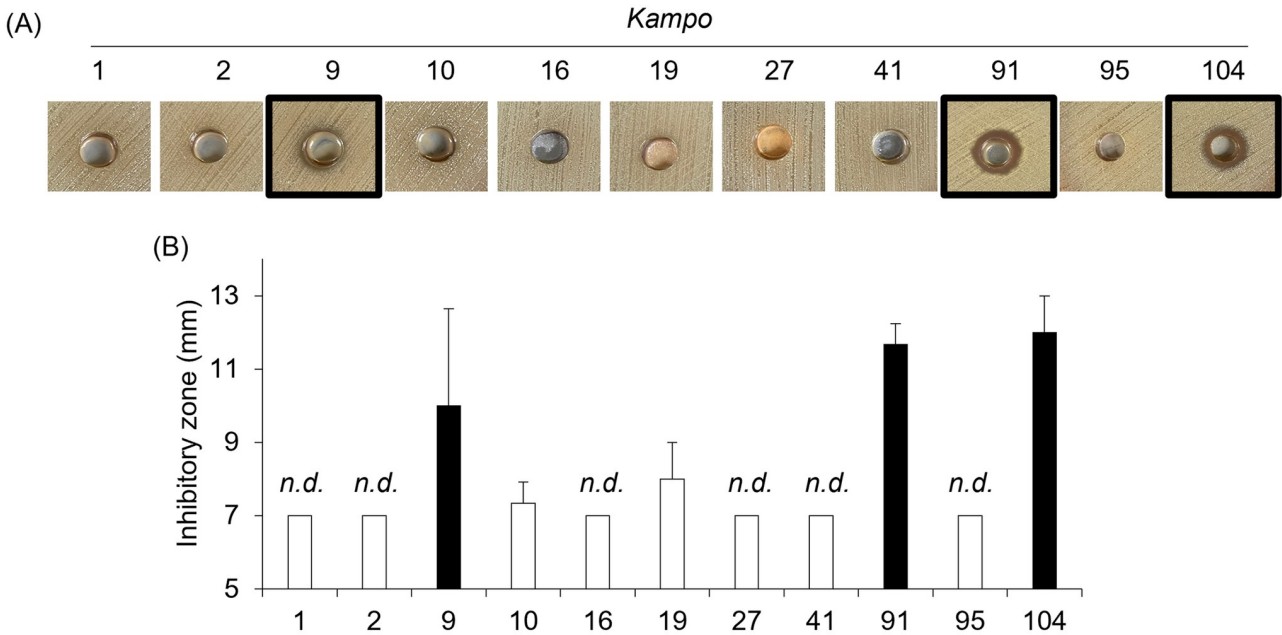

**Fig 2. The growth of *S. pneumoniae* serotype 1 is inhibited by treatment with Sho-saiko-To (9), Chikujo-untan-To (91), and Shin'i-seihai-To (104).** *S. pneumoniae* serotype 1 (A 0.5 McFarland suspension) was incubated on Mueller-Hinton II Agar containing 5% sheep blood in the presence of each *Kampo* product at 35˚C with 5% $CO_2$ overnight. (A) The representative photograph of the growth inhibition zone on Muller-Hinton Agar plates. (B) The growth inhibition zone around the hole was measured. Data are mean ± SD. Data from three independent experiments were combined. The white bar indicates that no inhibition was observed. n.d.: not detected.

and 2B). The growth of *S. pneumoniae* serotype 1 was suppressed by treatment with Sho-saiko-To (9), Chikujo-untan-To (91), and Shin'i-seihai-To (104).

## Bactericidal activity of Sho-saiko-To (9), Chikujo-untan-To (91) and Shin'i-seihai-To (104) against *S. pneumoniae* serotype 1 and 3

We then evaluated the MIC and MBC values of Sho-saiko-To (9), Sho-seiryu-To (91), and Shin'i-seihai-To (104) against *S. pneumoniae* serotype 1 and 3. The three selected *Kampo* products showed antibacterial activity against the strains, with MICs of 6.25–50 mg/mL, respectively (Table 1). Also, MBC of Sho-saiko-To (9), Sho-seiryu-To (91), and Shin'i-seihai-To (104) against *S. pneumoniae* serotypes 1 and 3 were 12.5–25 mg/mL.

## The growth of antibiotic-resistance *S. pneumoniae* strains is inhibited by treatment with Sho-saiko-To (9), Chikujo-untan-To (91), and Shin'i-seihai-To (104)

We then examined the antibacterial effects of selected four *Kampo* products, Sho-saiko-To (9), Sho-seiryu-To (19), Chikujo-untan-To (91), and Shin'i-seihai-To (104), on clinically isolated,

**Table 1. MIC and MBC of Sho-saiko-To (9), Chikujo-untan-To (91), and Shin'i-seihai-To (104) against *S. pneumoniae* serotype 1 and 3.**

|  | MIC (mg/mL) | | | MBC (mg/mL) | | |
|---|---|---|---|---|---|---|
|  | **9** | **91** | **104** | **9** | **91** | **104** |
| ***S. pneumoniae* serotype 1** | 50 | 50 | 12.5 | 25 | 25 | 25 |
| ***S. pneumoniae* serotype 3** | 6.25 | 12.5 | 12.5 | 12.5 | 12.5 | 25 |

**Table 2. Resistance of *S. pneumoniae* serotype 6A and 19A against TFLX.**

| Serotype | TFLX Inhibitory zone (mm) | Interpretive category |
|---|---|---|
| 6A | 6 | R |
| 19A | 20 | I |

I: Intermediate (The growth is inhibited *in vitro* by a concentration of this drug that is associated with an uncertain therapeutic effect), R: Resistant.

antibiotic-resistant *S. pneumoniae* serotype 6A and 19A. We first characterized the antibiotic resistance profiles of the clinical strains using TFLX, penicillin (PCG), meropenem (MEPM), erythromycin (EM), cefotaxime (CTX), imipenem (IPM), clindamycin (CM), and levofloxacin (LVFX). While *S. pneumoniae* serotype 6A was resistant to TFLX, EM, and LVFX, serotype 19A was resistant to EM and CLDM (Tables 2 and 3). We then treated *S. pneumoniae* 6A and 19A with four *Kampo* products. The growth of *S. pneumoniae* serotype 6A was inhibited by treatment with Sho-saiko-To (9), Chikujo-untan-To (91), and Shin'i-seihai-To (104), but not Sho-seiryu-To (19) whereas that of *S. pneumoniae* serotype 19A was suppressed by treatment with those products (Fig 3A–3D).

## The growth of *S. aureus* ATCC 43300 is inhibited by treatment with Sho-saiko-To (9) and Shin'i-seihai-To (104)

Then, we examined the antibacterial activity of 11 *Kampo* products against *S. aureus* ATCC 43300 (gram-positive), *K. pneumoniae* ATCC 700603 (gram-negative), and *E. coli* ATCC 25922 (gram-negative). The growth of *S. aureus* ATCC 43300 was inhibited by treatment with Sho-saiko-To (9) and Shin'i-seihai-To (104) (Fig 4A and 4B). Treatment with Hange-koboku-To (16) did not inhibit the growth of *S. aureus* ATCC 43300 in two out of three tests under the experimental condition. On the other hand, all *Kampo* products had no inhibitory activity against *K. pneumoniae* ATCC 700603 and *E. coli* ATCC 25922.

We then evaluated the MIC and MBC values of Sho-saiko-To (9) and Shin'i-seihai-To (104) against *S. aureus* ATCC 43300, *K. pneumoniae* ATCC 700603, and *E. coli* ATCC 25922. MIC of Sho-saiko-To (9) and Shin'i-seihai-To (104) for *S. aureus* ATCC 43300 was 12.5 mg/mL and 6.25 mg/mL, respectively (Table 4). Sho-saiko-To (9) and Shin'i-seihai-To (104) did not show bactericidal activity for *S. aureus* ATCC 43300, *K. pneumoniae* ATCC 700603, and *E. coli* ATCC 25922 in the experimental concentrations.

**Table 3. MIC of antibiotics against *S. pneumoniae* serotype 6A and 19A.**

| Antibiotics | Serotype 6A (μg/mL) | Serotype 19A (μg/mL) |
|---|---|---|
| PCG | 2.0 (S) | $\leqq$0.06 (S) |
| MEPM | 0.5 (I) | $\leqq$0.12 (S) |
| EM | 2.0 (R) | $\geqq$4.0 (R) |
| CTX | 1.0 (S) | $\leqq$0.5 (S) |
| IPM | $\leqq$0.12 (S) | $\leqq$0.12 (S) |
| CLDM | $\leqq$0.12 (S) | $\geqq$8.0 (R) |
| LVFX | $\geqq$16.0 (R) | 1.0 (S) |
| Remarks | | mefA+ermB gPISP(pbp2x) |

S: Susceptible, I: Intermediate (The growth is inhibited *in vitro* by a concentration of this drug that is associated with an uncertain therapeutic effect), R: Resistant.

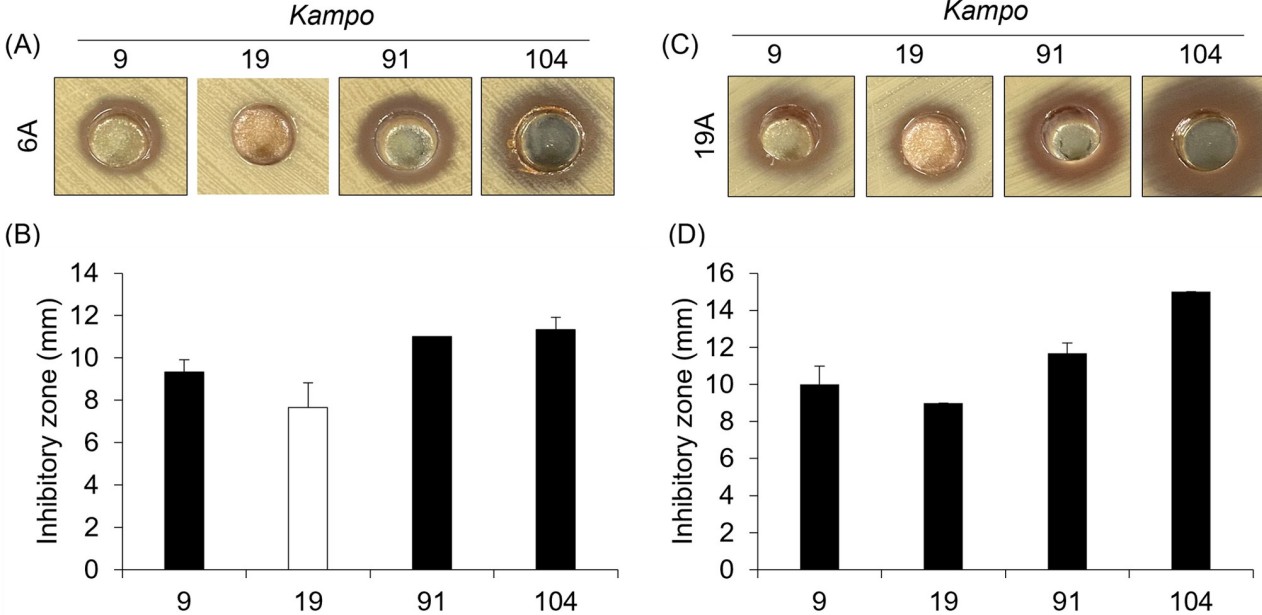

**Fig 3. The growth of *S. pneumoniae* serotypes 6A and 19A is inhibited by treatment with Sho-saiko-To (9), Chikujo-untan-To (91), and Shin'i-seihai-To (104).** *S. pneumoniae* serotype 6A and 19A (A 0.5 McFarland suspension) were incubated on Mueller-Hinton II Agar containing 5% sheep blood in the presence or absence of each *Kampo* product at 35°C with 5% $CO_2$ overnight. (A and C) The representative photograph of the growth inhibition zone on Muller-Hinton II Agar plates of *S. pneumoniae* serotype 6A and 19A. (B and D) The growth inhibition zones around the hole were measured (A and C). Data are mean ± SD. Data from three independent experiments were combined. S: Susceptible, I: Intermediate, R: resistant. The white bar indicates that no inhibition was observed.

## Discussion

In this study, we revealed the direct antibacterial effects of 11 *Kampo* products against *S. pneumoniae*. Notably, Sho-saiko-To (9), Sho-seiryu-To (19), Chikujo-untan-To (91), and Shin'i-seihai-To (104) exhibited significant growth inhibition activity against *S. pneumoniae* serotype 3, while Sho-saiko-To (9), Chikujo-untan-To (91), and Shin'i-seihai-To (104) effectively suppressed the growth of *S. pneumoniae* serotype 1. Furthermore, Sho-saiko-To (9), Chikujo-untan-To (91), and Shin'i-seihai-To (104) exerted antibacterial effects against antibiotic-resistant *S. pneumoniae* serotypes 6A and 19A. These suggest their potential as therapeutic agents for pneumococcal pneumonia treatment.

Additionally, Sho-saiko-To (9) and Shin'i-seihai-To (104) exhibited growth inhibition activity against *S. aureus* ATCC 43300, further expanding the spectrum of gram-positive bacteria affected by *Kampo* products. However, despite the promising effects against *S. pneumoniae* and *S. aureus* ATCC 43300, the *Kampo* products showed no inhibitory activity against *K. pneumoniae* ATCC 700603 and *E. coli* ATCC 25922. This suggests the possibility that gram-negative bacteria are not affected by the inhibitory action of Sho-saiko-To (9) and Shin'i-seihai-To (104). Gram-positive and gram-negative bacteria differ in the composition of the cell membrane structure. Lipopolysaccharide (LPS) is a different component between gram-positive and gram-negative bacteria and is suggested to function as a permeability barrier against antibiotics [37, 38]. Previously, Fukamachi et al. reported that Hange-Shashin-To (14) has growth inhibition activity for gram-negative bacteria, *Prevotella* spp., and *Porphyromonas* spp. [39]. Also, they showed Hange-shashin-To (14) had no antibacterial effect on *E. coli* [39]. It is likely that LPS is a factor uninvolved in the antibacterial effects of *Kampo* products on the

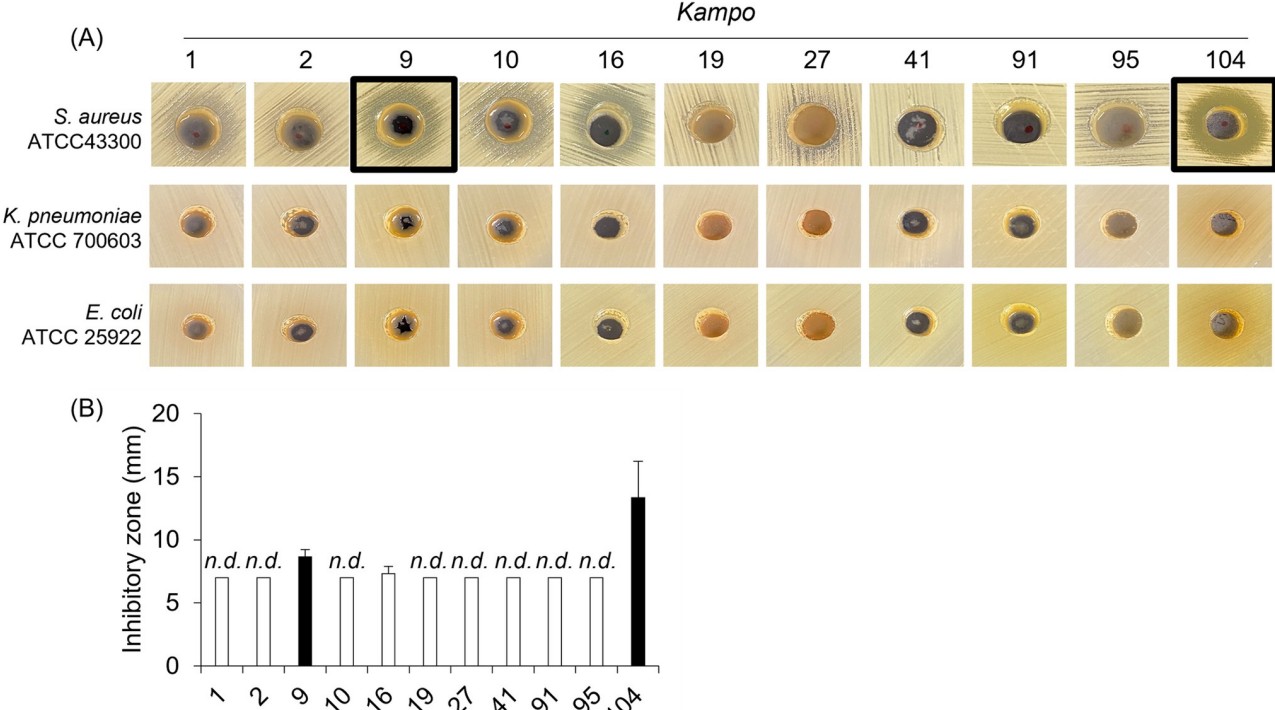

**Fig 4. The growth of *S. aureus* ATCC 43300 is inhibited by treatment with Sho-saiko-To (9) and Shin'i-seihai-To (104).** Mueller-Hinton II Agar was inoculated with 0.5 McFarland suspension of each bacterium. A hole was made in the agar with a cork borer. *Kampo* product was added dropwise to the spot. The plate was incubated at 35°C overnight. (A) The representative photograph of the growth inhibition zone on Muller-Hinton II Agar plates. (B) The growth inhibition zone around the hole was measured. Data are mean ± SD. Data from three independent assays were combined. The white bar indicates that no inhibition was observed. n.d.: not detected.

susceptibility of gram-negative bacteria. On the other hand, *E. coli* and *K. pneumoniae* have ACR family transporters, efflux proteins that belong to the resistance-nodulation-division (RND) family [40, 41]. It was reported that RND family transporters promote the active efflux of certain antibiotics [42]. In this context, ACR family transporters might contribute to the efflux of specific *Kampo* products, thereby leading to resistance of *E. coli* and *K. pneumoniae*.

Several studies indicated that Shin'i-seihai-To (104) had effects on *S. pneumoniae*. Minami et al. reported that Shin'i-seihai-To (104) had the ability to inhibit the biofilm formation of *S. pneumoniae* serotype 19F *in vitro* [24]. They also reported that the development of *S. pneumoniae* serotype-induced sinusitis in mice was suppressed by treatment with Shin'i-seihai-To (104) [43]. In this study, we clarified that Shin'i-seihai-To (104) exerts direct antibacterial effects on various strains of *S. pneumoniae*. Taken together, it is possible that Shin'i-seihai-To

**Table 4. MIC and MBC values of Sho-saiko-To (9) and Shin'i-seihai-To (104) against *S. aureus* ATCC 43300, *K. pneumoniae* ATCC 700603, and *E. coli* ATCC 25922.**

|  | MIC (mg/mL) | | MBC (mg/mL) | |
|---|---|---|---|---|
|  | 9 | 104 | 9 | 104 |
| *S. aureus* ATCC 43300 | 12.5 | 6.25 | >100 | >100 |
| *K. pneumoniae* ATCC 700603 | 100 | 25 | >100 | >100 |
| *E. coli* ATCC 25922 | 50 | 50 | >100 | >100 |

(104) has a significant potential for the treatment of pneumonia caused by *S. pneumoniae* infection.

Furthermore, we revealed the antibacterial effects of Sho-saiko-To (9) and Chikujo-untan-To (91) across a spectrum of *S. pneumoniae* strains. While the effector mechanisms and active compounds in each *Kampo* product may vary [44–46], these *Kampo* products present promise as therapies against pneumococcal pneumonia.

Several studies reported antibacterial efficacy tests of *Kampo* products against bacteria. Fukamachi et al. reported MIC values of Hange-shashin-To (177) against oral bacteria are 0.625–10 mg/mL [39]. Higaki et al. reported MIC values of 10 *Kampo* products against three bacteria (*Propionibacterium acnes*, *S. epidermidis*, and *S. aureus*) are 25–400 mg/mL [47, 48]. Consistent with these reports, MIC values of Sho-saiko-To (9) and Shin'i-seihai-To (104) against *S. pneumoniae* and *S. aureus* were 6.25–50 mg/mL and 6.25–12.5 mg/mL, respectively. From a pharmacokinetic viewpoint and clinical relevance, however, low-dose administration of *Kampo* products for the treatment of infectious diseases by *S. pneumoniae* and *S. aureus* might be preferable in terms of achievable concentrations in blood and local lung tissue. Basically, *Kampo* products consist of a combination of multiple crude drugs, and the crude drugs or their extracts are likely to have more superiority in drug efficacy. Liao et al. reported that Dai-ou and its extract, Aloe-emodin, suppress the growth of *Porphyromonas gingivalis* with MIC values of 500 µg/mL and 0.78 µg/mL, respectively [49]. Fukamachi et al. reported that the MIC values of Coptisine against oral bacteria are 4.4–35.6 µg/mL [39]. It is likely that crude drugs/extracts are therapeutically effective in treating infectious diseases. In this regard, we explored crude drugs in Sho-saiko-To (9) and Shin'i-seihai-To (104) and found Ou-gon as a common crude drug candidate capable of killing *S. pneumoniae* and *S. aureus*. Ou-gon has been reported to exhibit anti-fungal activity, such as *T. rubrum*, *Aspergillus fumigatus*, and *C. albicans* in 31–62 µg/mL [32, 50]. Thus, Ou-gon might have an antibacterial effect on *S. pneumoniae* and *S. aureus* at a similar dosage, whereas Ou-gon efficacy might be restricted by the LPS-containing membrane barrier in gram-negative bacteria. Also, Ou-gon contains flavonoids including Baicalin, Baicalein, and Wogonin. Baicalin and Baicalein have been reported to inhibit bacteria growth by targeting cell wall synthesis and inhibiting DNA replication [51–53]. Furthermore, these compounds have been reported to exhibit various physiological activities in animal experiments, such as anti-oxidant, anti-tumor, and anti-inflammatory activities [54–56]. In some cases, however, treatment with Ou-gon and Baicalin have been suggested to cause side effects such as interstitial pneumonitis and liver dysfunction [57, 58]. Therefore, it is considered that Ou-gon treatment should be utilized based on an understanding of the side effects.

In conclusion, we have shown the potential of *Kampo* products as a powerful alternative or adjunctive therapy against pneumonia-causative bacteria. With broad-spectrum antibacterial effects against *S. pneumoniae* and *S. aureus* ATCC 43300, *Kampo* products such as Sho-saiko-To (9) and Shin'i-seihai-To (104) may offer promise in addressing the challenges of pneumonia and antibiotic resistance.

## Supporting information

**S1 Table. Inhibitory zone test against TFLX.**
(PPTX)

**S2 Table. MIC value of serotypes 1 and 3.**
(PPTX)

**S3 Table. MBC value of serotypes 1 and 3.**
(PPTX)

**S1 Dataset. The dataset of Fig 1B.**
(PPTX)

**S2 Dataset. The dataset of Fig 2B.**
(PPTX)

**S3 Dataset. The dataset of Fig 3B and 3D.**
(PPTX)

**S4 Dataset. The dataset of Fig 4B.**
(PPTX)

## Acknowledgments

We acknowledge Kazuyoshi Kawakami, M.D. Ph.D., for kindly giving clinically isolated serotype 3 *Streptococcus pneumoniae*.

## Author Contributions

**Conceptualization:** Yukiko Akahori.

**Data curation:** Yukiko Akahori, Kenichi Shizuno.

**Formal analysis:** Yukiko Akahori, Yusuke Hashimoto.

**Funding acquisition:** Yukiko Akahori.

**Investigation:** Yukiko Akahori, Yusuke Hashimoto.

**Methodology:** Yukiko Akahori, Kenichi Shizuno.

**Supervision:** Mitsuaki Nagasawa.

**Validation:** Kenichi Shizuno.

**Writing – original draft:** Yukiko Akahori.

**Writing – review & editing:** Yukiko Akahori, Yusuke Hashimoto, Kenichi Shizuno, Mitsuaki Nagasawa.

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
