## [Decision Letter · Decision Letter 0]

5 Jul 2024

PONE-D-24-23283Antibacterial effects of Kampo medicines products against pneumonia causative bacteriaPLOS ONE

Dear Dr.  AKAHORI,

Thank you for submitting your manuscript to PLOS ONE. After careful consideration, we feel that it has merit but does not fully meet PLOS ONE’s publication criteria as it currently stands. Therefore, we invite you to submit a revised version of the manuscript that addresses the points raised during the review process.

We look forward to receiving your revised manuscript.

Kind regards,

Masaki Mogi

Academic Editor

PLOS ONE

Journal Requirements:

   "KAKENHI Grant-in-Aid for Early-Career Scientists（21K15654）Takeda Science Foundation"

Additional Editor Comments:

Two Reviewers well assessed the present papaer.However, there are several major critiques raised by the Reviewers.

Check the comments and respond them appropriately.

Reviewers' comments:

Reviewer's Responses to Questions

**Comments to the Author**

1. Is the manuscript technically sound, and do the data support the conclusions?

Reviewer #1: Yes

Reviewer #2: Partly

2. Has the statistical analysis been performed appropriately and rigorously? 

Reviewer #1: N/A

Reviewer #2: Yes

3. Have the authors made all data underlying the findings in their manuscript fully available?

Reviewer #1: Yes

Reviewer #2: Yes

4. Is the manuscript presented in an intelligible fashion and written in standard English?

Reviewer #1: Yes

Reviewer #2: Yes

5. Review Comments to the Author

Reviewer #1: Remarks to the authors

In this paper, the authors examined the direct anti-bacterial effects of 11 Kampo products on highly-virulent, pneumonia-causative bacteria commonly detected worldwide. The authors revealed that some Kampo products had bactericidal effects on highly-virulent Streptococcus pneumoniae strains. Of these, Kampo products (9), (91) and (104) exerted anti-bacterial effects on antibiotic-resistant S. pneumoniae strains. Furthermore, the author demonstrated that Kampo products (9) and (104) had inhibitory effects on the growth of Staphylococcus aureus. The present work is interesting. The manuscript might contribute to making a progress on the medical therapy for patients with pneumonia, but the author might consider the below comments to improve the manuscript, so as to be solid findings.

Comments/concerns

1. In this study, they demonstrated that several Kampo products are effective against S. pneumoniae and S. aureus, which are Gram-positive bacteria. However, the Kampo products did not show effectiveness against K. pneumoniae and E. coli, both that are Gram-negative bacteria. I think it is important to discuss that difference in effectiveness of these Kampo products against Gram-negative and Gram-positive bacteria in terms of components, structure or other relevant factors of Kampo products.

2. In diffusion assay, they showed no inhibitory effects of Kampo products including (9) and (104) on K. pneumoniae and E. coli (Figure 4A and B). On the contrary, they provided MIC values of (9) and (104) against the both bacteria by another assay (Table 4). Are the data consistent with each other?

Minor points

Title: Kampo medicines products -> Kampo products

LINE161 and 164: [91] -> (91)

Figure 1B: A black bar corresponding to 10 -> A white bar

Figure 2B: Black bars corresponding to 10 and 19 -> White bars

Figure 3B: A black bar corresponding to 19 -> A white bar. Add an explanation in Figure legend.

Figure 4B: A black bar corresponding to 16 -> A white bar

Reviewer #2: Reviewer Comments

The use of Kampo medicine in general medical practice, particularly for apparent bacterial infections, is not common. However, it is often practiced more with likely viral infectious conditions, and this fact is widely accepted across specialties. Additionally, the global issue of antimicrobial resistance among commensal bacteria, the monitoring of broad-spectrum antibiotic use regionally, and public awareness campaigns have been significant medical and societal concerns for over 20 years.

Researchers have undoubtedly focused on the direct antimicrobial effects of Kampo medicine, its effects on mucosal and immune cell activation, and the short-term elevation of blood concentrations. Given this, it is anticipated that many reviewers and readers will question how the fundamental pharmacokinetic properties of conventional antibiotics, which determine their efficacy particularly in alveolar regions where capillaries and the lower respiratory tract meet, are addressed in this study. This aspect is crucial and should not be overlooked, irrespective of the reader's specialty.

The core of this paper lacks a discussion on the pharmacokinetics of Kampo medicine. The presented MIC and MBC values of Kampo medicine, which are often in concentrations three orders of magnitude higher than typical antibiotics, need thorough explanation, especially regarding their clinical applicability for pneumonia treatment. This was a significant topic of discussion at a recent Japanese Society for Infectious Diseases conference, where the antimicrobial effect of Kampo on macrolide-resistant Streptococcus strains was presented. The presenter clarified that the intended use was for the direct inhibition of pharyngeal colonization by Streptococci, distinguishing it from the effectiveness in systemic organ infection.

Methods for measuring antimicrobial activity against clinical isolates, standardized by bodies such as CLSI and NCCLS, have been refined over the years. These methods have become benchmarks for clinical treatment efficacy, as the concentrations measured in micrograms per milliliter generally correlate with effective therapeutic doses.

Major Assessment

This paper reports the MIC and MBC of Kampo medicines against common causative bacteria in respiratory tract and soft tissue infections, including clinical isolates with notable drug resistance, using CLSI-standardized microdilution methods. The results are presented clearly.

However, as discussed, the inhibitory concentrations for pneumonia treatment need careful consideration regarding pharmacokinetics and clinical relevance. The paper should provide a detailed explanation of the gap between the MIC/MBC values and achievable blood/tissue concentrations. Moreover, the paper lacks discussion on which chemical components or combinations in the Kampo medicine might inhibit bacterial growth, which should be addressed if hypotheses exist.

Minor Recommendations

L16, p2: The term "biological effect" should be revised to "antimicrobial effect" since the study does not investigate host bioavailability or cellular biology.

L16, p2: Add a description of how common respiratory pathogens can colonize the upper respiratory tract or oral cavity and invade the lower respiratory tract, causing pneumonia.

L43, p3: Cite studies indicating the decreasing trend of community-acquired infections by PRSP and quinolone-resistant pneumococci over the past 25 years with one or two appropriate references and discuss whether this trend is likely to continue but concisely.

Introduction: While it is crucial to note that Kampo medicine exhibits antimicrobial effects, the mechanisms should be distinguished clearly between protozoa, viruses, bacteria, fungi, etc. The physiological non-realistic high concentrations only achievable on mucosal surfaces versus the pharmacokinetically feasible concentrations in the blood and target organs should be explicitly discussed.

L76, p5: Provide a brief explanation for using Todd-Hewitt broth for pneumococci versus nutrient-rich media for other bacteria, particularly regarding the susceptibility and growth phase considerations. For example, considering the fragility or unstable viability of pneumococcal live microbes on a simple agar surface.

Pneumococci MIC and MBC Testing: Were standard strains, such as ATCC, included in the parallel testing, and were MIC and MBC values measured for Serotype 3 pneumococci against antibiotics?

L93, p5: Clarify the volume of the bacterial suspension and the agar used in the growth inhibitory assay.

Figure 4B: Ensure the inhibitory effect or lack thereof for Kampo No. 16 is clearly interpreted and explained.

Overall Structure

The inclusion of figure legends within the results section should be reviewed for adherence to academic journal standards and, if necessary, addressed by the editorial team.

This version refines your original content for clarity and consistency, ensuring that the academic tone and structure are maintained throughout.

6. PLOS authors have the option to publish the peer review history of their article (what does this mean?). If published, this will include your full peer review and any attached files.

Reviewer #1: No

Reviewer #2: **Yes: **Natsuo Yamamoto, PhD, MD

---

## [Author Response · Author response to Decision Letter 0]

16 Aug 2024

Reviewers’ comments:

Reviewer #1:

Thank you very much for your time and the precious comments. We describe the responses to the comments below.

1. In this study, they demonstrated that several Kampo products are effective against S. pneumoniae and S. aureus, which are Gram-positive bacteria. However, the Kampo products did not show effectiveness against K. pneumoniae and E. coli, both that are Gram-negative bacteria. I think it is important to discuss that difference in effectiveness of these Kampo products against Gram-negative and Gram-positive bacteria in terms of components, structure or other relevant factors of Kampo products.

Response: Thank you for the important comment. According to the reviewer’s suggestion, we examined components and structures from selected Kampo products. However, we could not find general structural features of the Kampo components. Notably, we discussed why E. coli and K. pneumoniae were resistant to the Kampo products. We added the description in the Discussion section (L260-272, p15-16).

2. In diffusion assay, they showed no inhibitory effects of Kampo products including (9) and (104) on K. pneumoniae and E. coli (Figure 4A and B). On the contrary, they provided MIC values of (9) and (104) against the both bacteria by another assay (Table 4). Are the data consistent with each other?

Response: The data are consistent. In the test guideline by CLSI, a 0.5 McFarland bacteria suspension is used in both assays. These methods are quite different; in diffusion assay, the bacterial suspension is swabbed to agar plates, and each Kampo suspension is dropped into the prepared hole on the agar plate. For evaluation of MIC, the bacterial suspension (50 �L) is applied to serially diluted Kampo suspensions in 96-well plates. Therefore, the number of bacteria and the amount of Kampo suspensions are not equal between both assays.

Minor points

Title: Kampo medicines products -> Kampo products

LINE161 and 164: [91] -> (91)

Figure 1B: A black bar corresponding to 10 -> A white bar

Figure 2B: Black bars corresponding to 10 and 19 -> White bars

Figure 3B: A black bar corresponding to 19 -> A white bar. Add an explanation in Figure legend.

Figure 4B: A black bar corresponding to 16 -> A white bar

Response: We have corrected all the minor points you have pointed out.

Reviewer #2:

Thank you very much for your time and the important comments. We describe the responses to the comments below.

Major Assessment

This paper reports the MIC and MBC of Kampo medicines against common causative bacteria in respiratory tract and soft tissue infections, including clinical isolates with notable drug resistance, using CLSI-standardized microdilution methods. The results are presented clearly.

However, as discussed, the inhibitory concentrations for pneumonia treatment need careful consideration regarding pharmacokinetics and clinical relevance. The paper should provide a detailed explanation of the gap between the MIC/MBC values and achievable blood/tissue concentrations. Moreover, the paper lacks discussion on which chemical components or combinations in the Kampo medicine might inhibit bacterial growth, which should be addressed if hypotheses exist.

Response: Thank you for the critical comment. In this study, we showed the inhibitory concentrations of Kampo products against S. pneumoniae and S. aureus tended to be relatively high. The level of inhibitory effect is considered unpreferable for treating pneumonia caused by respiratory bacteria. On the other hand, Kampo products contain crude drugs, which are known to be effective against some bacteria even at low concentrations (Liao et al., Pharm. Biol., 2013; Fukamachi et al., Evid. Based Complement. Alternat. Med., 2015). We explored crude drugs in Sho-saiko-To (9) and Shin'i-seihai-To (104) and found that the Kampo products contained Ou-gon as a common crude drug. Ou-gon has known to have antifungal properties (Da et al., Nat. Prod. Commun., 2016; Da et al., Sci. Rep., 2019). Therefore, it is suggested that Ou-gon might have the potential to exhibit antimicrobial activity against S. pneumoniae and S. aureus at low concentrations. We added the description in the Discussion section (L284-303, p16-17).

Minor Recommendations

L16, p2: The term "biological effect" should be revised to "antimicrobial effect" since the study does not investigate host bioavailability or cellular biology.

Response: Thank you for the comment. We replaced the term in the Abstract section (L19, p2).

L16, p2: Add a description of how common respiratory pathogens can colonize the upper respiratory tract or oral cavity and invade the lower respiratory tract, causing pneumonia.

Response: We added the description in the Introduction section (L39-43, p3).

L43, p3: Cite studies indicating the decreasing trend of community-acquired infections by PRSP and quinolone-resistant pneumococci over the past 25 years with one or two appropriate references and discuss whether this trend is likely to continue but concisely.

Response: We added the description in the Introduction section (L47-54, p3).

Introduction: While it is crucial to note that Kampo medicine exhibits antimicrobial effects, the mechanisms should be distinguished clearly between protozoa, viruses, bacteria, fungi, etc. The physiological non-realistic high concentrations only achievable on mucosal surfaces versus the pharmacokinetically feasible concentrations in the blood and target organs should be explicitly discussed.

Response: We have rewritten the text in the Introduction section in detail (L66-75, p4). Also, we added the description about the pharmacokinetics of Kampo products in the Discussion section (L284-303, p16-17).

L76, p5: Provide a brief explanation for using Todd-Hewitt broth for pneumococci versus nutrient-rich media for other bacteria, particularly regarding the susceptibility and growth phase considerations. For example, considering the fragility or unstable viability of pneumococcal live microbes on a simple agar surface.

Response: We added the description in the Materials and Methods section (L90-91, p6).

Pneumococci MIC and MBC Testing: Were standard strains, such as ATCC, included in the parallel testing, and were MIC and MBC values measured for Serotype 3 pneumococci against antibiotics?

Response: We used the clinically isolated strains only in this study. According to your comments, we tested the MIC and MBC values of serotype 1 and serotype 3. The data are added in Supplementary information as S1-3 Table.

L93, p5: Clarify the volume of the bacterial suspension and the agar used in the growth inhibitory assay.

Response: According to the test guideline by CLSI, we have inoculated the bacteria using a swab. We corrected the description in the Materials and Methods section (L107-109, p6-7).

Figure 4B: Ensure the inhibitory effect or lack thereof for Kampo No. 16 is clearly interpreted and explained.

Response: Kampo No. 16 lacks the inhibitory effect. We have corrected the Figure 4B.

Overall Structure

The inclusion of figure legends within the results section should be reviewed for adherence to academic journal standards and, if necessary, addressed by the editorial team.

Response: We have written the manuscript in accordance with the guidelines provided by PLOS ONE.

---

## [Decision Letter · Decision Letter 1]

4 Sep 2024

PONE-D-24-23283R1Antibacterial effects of Kampo products against pneumonia causative bacteriaPLOS ONE

Dear Dr. AKAHORI,

Thank you for submitting your manuscript to PLOS ONE. After careful consideration, we feel that it has merit but does not fully meet PLOS ONE’s publication criteria as it currently stands. Therefore, we invite you to submit a revised version of the manuscript that addresses the points raised during the review process.

The manuscript has been improved. Reviewer #2's comments are constructive to improve the manuscript better. See the comments and repond them appropriately.

We look forward to receiving your revised manuscript.

Kind regards,

Masaki Mogi

Academic Editor

PLOS ONE

Journal Requirements:

Reviewers' comments:

Reviewer's Responses to Questions

**Comments to the Author**

1. If the authors have adequately addressed your comments raised in a previous round of review and you feel that this manuscript is now acceptable for publication, you may indicate that here to bypass the “Comments to the Author” section, enter your conflict of interest statement in the “Confidential to Editor” section, and submit your "Accept" recommendation.

Reviewer #1: All comments have been addressed

Reviewer #2: (No Response)

2. Is the manuscript technically sound, and do the data support the conclusions?

Reviewer #1: Yes

Reviewer #2: Partly

3. Has the statistical analysis been performed appropriately and rigorously? 

Reviewer #1: N/A

Reviewer #2: Yes

4. Have the authors made all data underlying the findings in their manuscript fully available?

Reviewer #1: Yes

Reviewer #2: Yes

5. Is the manuscript presented in an intelligible fashion and written in standard English?

Reviewer #1: Yes

Reviewer #2: Yes

6. Review Comments to the Author

Reviewer #1: (No Response)

Reviewer #2: Reviewer Comments:

The revised manuscript shows considerable improvement, particularly in addressing the complexity of Kampo compounds’ composition, its antimicrobial effects, and the standardized quantification of MIC and MBC values. The rationale behind your conclusions is now more clearly articulated, and the previous concerns raised in the initial review have been adequately addressed. Given these enhancements, the manuscript is on a promising path towards acceptance.

Ｉｎ last addition, I would like to suggest a minor modification regarding the discussion on the limited efficacy of Kampo compounds against Gram-negative bacteria. Specifically, you mentioned the role of LPS (lipopolysaccharides) as a potential barrier. It would strengthen the manuscript if you could incorporate a few more relevant citations and consider the following explanation:

The LPS-rich outer membrane of Gram-negative bacteria is known to act as a barrier against many antimicrobial agents. Scutellaria baicalensis (Ougon) contains flavonoids such as baicalin and baicalein, which have been reported to inhibit bacterial growth, particularly in Gram-positive bacteria, by targeting cell wall synthesis and DNA replication. LPS-containing outer membrane may resist against Ougon incorporation. Additionally, these compounds exhibit various physiological activities within the host. Incorporating a brief discussion of these established effects, perhaps before the ACR family discussion on page 15, could further substantiate the study's relevance.

While it is hypothesized that the rise in blood concentrations of these lipophilic flavonoids, such as baicalin and baicalein, occurs relatively rapidly following standard oral dosing, their sufficiency in achieving therapeutic levels in the inflamed lung tissue remains uncertain. Current research on their in vivo antimicrobial activity is still limited, and this should be carefully examined in future studies. If this line of reasoning aligns with your findings, it might be worthwhile to include it in the discussion. However, given that your investigation focuses on the overall effects of the complex mixture within Kampo medicine, where flavonoids are not the sole active components, this addition is not mandatory.

7. PLOS authors have the option to publish the peer review history of their article (what does this mean?). If published, this will include your full peer review and any attached files.

Reviewer #1: No

Reviewer #2: **Yes: **Natsuo Yamamoto, PhD, MD.

---

## [Author Response · Author response to Decision Letter 1]

3 Oct 2024

Reviewers’ comments:

Reviewer #1:

Thank you very much for your time to review our article.

Reviewer #2:

Thank you very much for your time and the precious comments. We describe the responses to the comments below.

In last addition, I would like to suggest a minor modification regarding the discussion on the limited efficacy of Kampo compounds against Gram-negative bacteria. Specifically, you mentioned the role of LPS (lipopolysaccharides) as a potential barrier. It would strengthen the manuscript if you could incorporate a few more relevant citations and consider the following explanation:

The LPS-rich outer membrane of Gram-negative bacteria is known to act as a barrier against many antimicrobial agents. Scutellaria baicalensis (Ougon) contains flavonoids such as baicalin and baicalein, which have been reported to inhibit bacterial growth, particularly in Gram-positive bacteria, by targeting cell wall synthesis and DNA replication. LPS-containing outer membrane may resist against Ougon incorporation. Additionally, these compounds exhibit various physiological activities within the host. Incorporating a brief discussion of these established effects, perhaps before the ACR family discussion on page 15, could further substantiate the study's relevance.

Response: Thank you for your comment. We added the description in the Discussion section (L264 p.15, L301-302, L304-312 p.17). Also, we eliminated irrelevant descriptions from the text.

While it is hypothesized that the rise in blood concentrations of these lipophilic flavonoids, such as baicalin and baicalein, occurs relatively rapidly following standard oral dosing, their sufficiency in achieving therapeutic levels in the inflamed lung tissue remains uncertain. Current research on their in vivo antimicrobial activity is still limited, and this should be carefully examined in future studies. If this line of reasoning aligns with your findings, it might be worthwhile to include it in the discussion. However, given that your investigation focuses on the overall effects of the complex mixture within Kampo medicine, where flavonoids are not the sole active components, this addition is not mandatory.

Response: Thank you for your comment. The metabolism and blood concentration levels of some flavonoid compounds have been investigated in detail. Also, as per the reviewer’s comment, it is unclear whether the concentrations of flavonoids in inflamed lung tissues are sufficient for therapeutic levels. Furthermore, it is suggested the possibility that Baicalin exhibits side effects such as interstitial pneumonia and liver dysfunction. Therefore, I think that it is necessary to carefully study their in vivo activity. However, we did not describe the discussion by considering that the present study focused on the efficacy of Kampo products, which are a mixture of Shouyaku (crude drug), but not flavonoids, as described by the reviewer. I think that this is beyond the scope of discussion in the present study.

---

## [Decision Letter · Decision Letter 2]

8 Oct 2024

Antibacterial effects of Kampo products against pneumonia causative bacteria

PONE-D-24-23283R2

Dear Dr. Akahori,

We’re pleased to inform you that your manuscript has been judged scientifically suitable for publication and will be formally accepted for publication once it meets all outstanding technical requirements.

Kind regards,

Masaki Mogi

Academic Editor

PLOS ONE

Additional Editor Comments (optional):

No further comment.

Reviewers' comments:

Reviewer's Responses to Questions

**Comments to the Author**

1. If the authors have adequately addressed your comments raised in a previous round of review and you feel that this manuscript is now acceptable for publication, you may indicate that here to bypass the “Comments to the Author” section, enter your conflict of interest statement in the “Confidential to Editor” section, and submit your "Accept" recommendation.

Reviewer #2: All comments have been addressed

2. Is the manuscript technically sound, and do the data support the conclusions?

Reviewer #2: Partly

3. Has the statistical analysis been performed appropriately and rigorously? 

Reviewer #2: N/A

4. Have the authors made all data underlying the findings in their manuscript fully available?

Reviewer #2: Yes

5. Is the manuscript presented in an intelligible fashion and written in standard English?

Reviewer #2: Yes

6. Review Comments to the Author

Reviewer #2: (No Response)

7. PLOS authors have the option to publish the peer review history of their article (what does this mean?). If published, this will include your full peer review and any attached files.

Reviewer #2: **Yes: **Natsuo Yamamoto, PhD, MD

---

## [Editor Report · Acceptance letter]

17 Oct 2024

PONE-D-24-23283R2 

PLOS ONE

Dear Dr. AKAHORI, 

I'm pleased to inform you that your manuscript has been deemed suitable for publication in PLOS ONE. Congratulations! Your manuscript is now being handed over to our production team.

Kind regards, 

on behalf of

Dr. Masaki Mogi 

Academic Editor

PLOS ONE